# Overexpression of a Thermostable α-Amylase through Genome Integration in *Bacillus subtilis*

Yifan Yang [1,†], Xiaoping Fu [2,3,4,†], Xingya Zhao [2,3,4], Jianyong Xu [2,3,4], Yihan Liu [1,*], Hongchen Zheng [2,3,4,5,*], Wenqin Bai [2,3,4,5,*] and Hui Song [2,3,4]

1. Key Laboratory of Industrial Fermentation Microbiology, Ministry of Education, The College of Biotechnology, Tianjin University of Science and Technology, Tianjin 300457, China
2. National Center of Technology Innovation for Synthetic Biology, Tianjin Institute of Industrial Biotechnology, Chinese Academy of Sciences, Tianjin 300308, China
3. Industrial Enzymes National Engineering Research Center, Tianjin Institute of Industrial Biotechnology, Chinese Academy of Sciences, Tianjin 300308, China
4. Tianjin Key Laboratory for Industrial Biological Systems and Bioprocessing Engineering, Tianjin Institute of Industrial Biotechnology, Chinese Academy of Sciences, Tianjin 300308, China
5. Key Laboratory of Engineering Biology for Low-Carbon Manufacturing, Tianjin Institute of Industrial Biotechnology, Chinese Academy of Sciences, Tianjin 300308, China
* Correspondence: lyh@tust.edu.cn (Y.L.); zheng_hc@tib.cas.cn (H.Z.); baiwq@tib.cas.cn (W.B.); Tel.: +86-022-8486-1933 (H.Z.)
† These authors contributed equally to this work.

**Abstract:** A carbohydrate binding module 68 (CBM68) of pullulanase from *Anoxybacillus* sp. LM18-11 was used to enhance the secretory expression of a thermostable α-amylase (BLA702) in *Bacillus subtilis*, through an atypical secretion pathway. The extracellular activity of BLA702 guided by CBM68 was 1248 U/mL, which was 12.6 and 7.2 times higher than that of BLA702 guided by its original signal peptide and the endogenous signal peptide LipA, respectively. A single gene knockout strain library containing 51 genes encoding macromolecular transporters was constructed to detect the effect of each transporter on the secretory expression of CBM68-BLA702. The gene knockout strain 0127 increased the extracellular amylase activity by 2.5 times. On this basis, an engineered strain *B. subtilis* 0127 (AmyE::BLA702-NprB::CBM68-BLA702-PrsA) was constructed by integrating BLA702 and CBM68-BLA702 at the AmyE and NprB sites in the genome of *B. subtilis* 0127, respectively. The molecular chaperone PrsA was overexpressed, to reduce the inclusion body formation of the recombinant enzymes. The highest extracellular amylase activity produced by *B. subtilis* 0127 (AmyE::BLA702-NprB::CBM68-BLA702-PrsA) was 3745.7 U/mL, which was a little lower than that (3825.4 U/mL) of *B. subtilis* 0127 (pMAC68-BLA702), but showing a better stability of passage. This newly constructed strain has potential for the industrial production of BLA702.

**Keywords:** thermostable α-amylase; genome integration; *Bacillus subtilis*; CBM68; PrsA

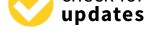



## 1. Introduction

*Bacillus subtilis* has long been used as a model organism for molecular research, as well as an industrial workhorse for the production of valuable enzymes [1,2]. Compared with other bacillus species, such as *Bacillus licheniformis* and *Bacillus amyloliquefaciens*, *B. subtilis*, with its more mature molecular operating systems and tools, has shown advantages in genetic regulation, enhancing the expression of heterologous proteins [3,4]. At present, using rational chassis cell engineering to construct microbial cell factories is widely regarded as an effective strategy for producing valuable industrial enzyme products [5–7]. Rational chassis engineering includes the deletion of undesirable intrinsic genes and negative regulation factors, the optimization of the protein secretion pathway, and the overexpression of molecular chaperones and so on [8]. The secretory expression systems of *B. subtilis* are a matter of concern in industrial applications and need to be further investigated

and developed [9]. To date, the majority of exported proteins are transported through the general secretory (Sec) pathway, in cooperation with the signal recognition particle (SRP) [10]. However, recent proteome analysis of *B. subtilis* extracellular proteins has revealed that approximately 50% of these proteins were not projected in previous genome-based predictions [11]. In recent years, more and more studies have reported that certain foreign proteins without signal peptides and specific motifs can be directly secreted into the medium through an atypical *B. subtilis* secretion pathway [9,11]. Moreover, molecular chaperone protein PrsA, as a lipoprotein attached to the outer part of the membrane, is essential for cell viability, due to its function in assisting the folding of the penicillin binding proteins responsible for the synthesis of the cell wall [12]. PrsA may also take on other roles and there is now good evidence that it acts as a very efficient folding catalyst for exported amylases and various other enzymes [12–14].

In contrast to the use of plasmids as genetic carriers, genome integration ensures that heterologous genetic material is stably maintained during passage and fermentation [15]. The natural genetic transformation system of *B. subtilis* facilitates the integration of DNA into the genome through homologous recombination [15,16]. Since DNA replication originates at a fixed position in the genome (origin of replication), and as rapidly growing bacteria initiate new rounds of replication before the previous rounds have been completed, the copy number of integrated genes fluctuates. As a result, gene dosage rises with increasing proximity to the origin of replication [17]. Another factor that might affect the expression level of integrated genes is functioning under different promoters. According to a previous report, there was a strong gene dosage effect in fast-growing *B. subtilis* cells, which could amount to a 5-fold difference in gene expression [15]. However, there have been few reports on strategies and methods for efficient expression of foreign proteins through genome integration.

Recent advances in systems biology have enabled analyses of multilayer omics, including transcriptome, proteome, metabolome, and macromolecular transporters, with use of statistical and computational models [2,18–20]. In this study, multiple novel and effective regulation strategies were developed in *B. subtilis* SCK6, to enhance the secretory expression of a highly thermostable amylase BLA702 from *Bacillus licheniformis* 702. The rational engineering of the production strain was based on a computational analysis using bioinformatics methods. Through optimized multiple rational engineering of the amylase production strain, the extracellular amylase activity was increased by 6.2 times. The final engineered genome integration strain, *B. subtilis* 0127 (AmyE::BLA702-NprB::CBM68-BLA702-PrsA), which has a high expression level and high passage stability, is a potential industrial production strain. The novel strategies used in this study could also facilitate basic research and biotechnological applications of *B. subtilis*.

## 2. Materials and Methods

### 2.1. Bacterial Strains, Plasmids, and Chemicals

*Bacillus subtilis* SCK6 was obtained from the Bacillus Genetic Stock Center (http://www.bgsc.org) with the accession code of 1A976 [21]. *Bacillus licheniformis* 702, which is the wild-type production strain of the highly thermostable amylase (BLA702), was deposited in our laboratory. The plasmids used in this study are listed in Table 1. pMA0911 was used as a vector for heterologous expression in *B. subtilis* SCK6. pDG1730 was used as a vector for genome integration in *B. subtilis* SCK6. The chemicals and solutions were of analytical reagent grade purity and obtained from commercial sources.

**Table 1.** Plasmids used in this study.

| Plasmids | Properties | Reference |
|---|---|---|
| pMA0911-SP-BLA702 | *E.coli*/*B. subtilis* shuttle plasmid, Ampr (*E. coli*), Kanr (*B. subtilis*), SP signal peptide, highly thermostable amylase gene | In this work |
| pMA0911-LipA-BLA702 | *E.coli*/*B. subtilis* shuttle plasmid, Ampr (*E. coli*), Kanr (*B. subtilis*), LipA signal peptide, highly thermostable amylase gene | In this work |
| pMA0911-LipA-BLA702codon | *E.coli*/*B. subtilis* shuttle plasmid, Ampr (*E. coli*), Kanr (*B. subtilis*), LipA signal peptide, Codon optimized gene sequence of BLA702 | In this work |
| pMA0911-YncM-BLA702 | *E.coli*/*B. subtilis* shuttle plasmid, Ampr (*E. coli*), Kanr (*B. subtilis*), YncM signal peptide, High thermostable amylase gene | In this work |
| pMA0911-CBM68-BLA702 | *E.coli*/*B. subtilis* shuttle plasmid, Ampr (*E. coli*), Kanr (*B. subtilis*), CBM68 domain, highly thermostable amylase gene | In this work |
| pMAC68-BLA702-0046 | *E.coli*/*B. subtilis* shuttle plasmid, Ampr (*E. coli*), Kanr (*B. subtilis*), CBM68-BLA702, *ytnA* gene | In this work |
| pMAC68-BLA702-0572 | *E.coli*/*B. subtilis* shuttle plasmid, Ampr (*E. coli*), Kanr (*B. subtilis*), CBM68-BLA702, *comEA* gene | In this work |
| pMAC68-BLA702-1521 | *E.coli*/*B. subtilis* shuttle plasmid, Ampr (*E. coli*), Kanr (*B. subtilis*), CBM68-BLA702, *motA* gene | In this work |
| pMAC68-BLA702-1951 | *E.coli*/*B. subtilis* shuttle plasmid, Ampr (*E. coli*), Kanr (*B. subtilis*), CBM68-BLA702, *crh* gene | In this work |
| pMAC68-BLA702-2143 | *E.coli*/*B. subtilis* shuttle plasmid, Ampr (*E. coli*), Kanr (*B. subtilis*), CBM68-BLA702, *SpoIIQ* gene | In this work |
| pDG1730-BLA702 | *B. subtilis* integrative plasmid, Ampr (*E. coli*), Spcr (*B. subtilis*), upstream and downstream sequences of homologous gene *amyE*, sandwiched between them is the BLA702 gene | In this work |
| pDG1730-nprB-BLA702/CBM68-BLA702 | *B. subtilis* integrative plasmid, Ampr (*E. coli*), Spcr (*B. subtilis*), upstream and downstream sequences of homologous gene *nprB*, sandwiched between them is the BLA702 or CBM68-BLA702 gene | In this work |
| pDG1730-aprE-PrsA | *B. subtilis* integrative plasmid, Ampr (*E. coli*), Spcr (*B. subtilis*), upstream and downstream sequences of homologous gene *aprE*, sandwiched between them is the chaperonin PrsA gene | In this work |

### 2.2. Construction of the Recombinant Plasmids and Strains

All recombinant plasmids for heterologous expression and genome integration are listed in Table 1. Schematic diagrams of the plasmid construction are shown in Figure S1. The open reading frame (ORF) of the highly thermostable amylase gene was amplified from the genome of *Bacillus licheniformis* 702 using the primers SP-F/SP-F (Table S1). The expression plasmid pMA0911-SP-BLA702 was constructed using the golden gate method [22]. Using the same method, the expression plasmids pMA0911-LipA-BLA702, pMA0911-LipA-BLA702codon, pMA0911-YncM-BLA702, and pMA0911-CBM68-BLA702 were constructed with the related primers listed in Table S1 and using pMA0911 as the skeleton. The plasmids that overexpressed the key transporter genes increasing the extracellular expression of BLA702 were constructed based on the plasmid skeleton of pMAC68-BLA702. The related genes *ytnA* (0046), *comEA* (0572), *motA* (1521), *crh* (1951), and *SpoIIQ* (2143) were amplified from chromosomal DNA of *B. subtilis* SCK6 using the related primer pairs listed in Table S1 and inserted right after the BLA702 gene. The genome integration plasmid pDG1730 was used to insert the target genes into the *amyE* gene site of the genome of *B. subtilis*. The recombinant plasmids pDG1730-nprB and pDG1730-aprE were constructed by replacing

the upstream and downstream sequences of *amyE* with the related homologous arms of *nprB* and *aprE*, respectively. The target genes encoding BLA702, CBM68-BLA702, and PrsA were inserted into the middle of the related homologous arms. All of the abovementioned expression plasmids and genome integration plasmids were transformed into competent cells of *B. subtilis* SCK6 or *B. subtilis* 0127 to obtain the related recombinant strains, according to the method described in our previous work [9]. The positive transformants were identified using bacteria liquid PCR with the test primers listed in Table S1. Then, sequencing confirmation followed, to obtain the recombinant strains.

### 2.3. Construction of the Single Gene Knockout Strain Library of Macromolecular Transporters

A macromolecular transporter analysis of *B. subtilis* SCK6 was performed using the online software SPLIT 4.0 SERVER (http://split.pmfst.hr/split/4/, accessed on 25 January 2021). A total of 51 macromolecular transporters were chosen to construct the single gene knockout strain library (Table S2). The strain library was constructed using a base editing method for genome editing in *B. subtilis* SCK6 and utilizing CRISPR/dCas9 [23]. All the N20s in this study were designed utilzing the website http://gbig.ibiodesign.net/index.html (accessed on 8 June 2021) and are listed in Table S3. The editing plasmids were constructed using a Golden Gate assembly reaction, with pBAC-dCas9-AID as the plasmid backbone [23]. *B. subtilis* SCK6 was transformed with 2 μg editing plasmids pBAC-dCas9-AID-gRNAtarget$^{TS}$ using the super-competent method [21,23]. Then the transformed cells were spread on LB agar plates supplemented with 10 μg/mL chloramphenicol and grown at 30 °C. Single colonies were selected and cultured at 30 °C overnight in LB medium containing 10 μg/mL chloramphenicol. The cultural broth was diluted to an initial $OD_{600nm}$ of 0.1 in 3 mL LB medium containing 10 μg/mL chloramphenicol and 0.1 mM IPTG. They were subsequently grown for a further 24 h at 30 °C. The targeted genomic regions were amplified using PCR from these cultures to verify the editing events through Sanger sequencing [23]. Thereafter, the strains containing edited plasmids were incubated at 37 °C for 24 h in antibiotic-free LB medium. Subsequently, the cultures were diluted and spread on LB with and without chloramphenicol. After incubation at 30 °C for 48 h, the single colonies that had been grown on LB plates without chloramphenicol were streaked on new LB plates and grown at 30 °C with and without antibiotics, to confirm plasmid loss [23].

### 2.4. Fermentation of the Engineered Strains to Produce Thermostable α-Amylase

Single colonies were selected after LB plate (25 μg/mL kanamycin) activation and inoculated into fresh LB culture liquid (25 μg/mL kanamycin) for 14 h of cultivation. The culture broth was then transferred to 30 mL SR medium (30 g/L tryptone, 50 g/L yeast Extract, and 6 g/L $K_2HPO_4$) with 1% inoculum in a 250 mL shake flask. It was cultured at 37 °C and 220 rpm for 48–168 h, to express the target enzymes. The extracellular amylase activity and protein expression were measured at fixed intervals. When the genome integration strains were fermented for enzyme production, no kanamycin supplementation was used.

### 2.5. Production Stability of the Engineered Strains

The optimum amylase production strains *B. subtilis* 0127 (pMAC68-BLA702) and *B. subtilis* 0127 (AmyE::BLA702-NprB::CBM68-BLA702-PrsA) were selected in this study, to further compare the strain stability after successive inoculations. The two engineered strains were successively inoculated for 10 generations without the addition of any antibiotics. The tenth-generation strains of the two engineered strains were fermented using the above method, to evaluate their enzyme production capacities. The first generation strains were used as the control.

*2.6. Determination of the Extracellular Amylase Activity and Protein Expression*

Extracellular products were collected to simultaneously determine the amylase activity and the expression of target proteins. The amylase activity was measured according to methods reported previously, with slight modifications [5,24,25]. One unit of amylase was defined as the amount of amylase needed per minute to complete the liquefaction of 1 mg starch into dextrin at pH 6.0 and 70 °C. The calculation of the enzyme activity was based on the formula X = c × n × 16.67, where X is the enzyme activity of the sample (U/mL); c is the concentration of the control enzyme (U/mL) corresponding to the absorbance, which was obtained from the appendix of the Standard (QB/T 2306-97); and n is the fold dilution [5,25]. Extracellular protein expression was measured using a sodium dodecyl sulfate polyacrylamide gel electrophoresis (SDS-PAGE) method reported previously [9].

*2.7. Data Analytic Methods*

Statistical analysis was carried out using GraphPad Prism 6.01 (GraphPad Software Inc., San Diego, CA, USA). The results are presented as the mean ± the standard deviation (SD) for a replication of *n* = 3. Analysis of variance (ANOVA) and comparison of the means were conducted with a multiple range comparison LSD test. A probability value of *p* < 0.05 was considered significant.

**3. Results**

*3.1. Fusing CBM68 to the N-Terminal of BLA702, Enhancing Its Secretory Expression in B. subtilis SCK6*

In our previous report, a pullulanase (PulA) from *Anoxybacillus* sp. LM18-11 was identified as being overexpressed in *B. subtilis* through a non-classical secretion pathway [9]. The N-terminal domain CBM68 may facilitate the secretory expression of the whole protein molecule. Thus, in this study, the CBM68 domain was used as a signal leader on the N-terminal of a thermostable α-amylase BLA702 from *B. licheniformis* 702 (Figure 1a). The recombinant fused protein CBM68-BLA702 showed similar a three-dimensional structure as that of PulA (Figure 1a). After 48 h fermentation, the extracellular amylase activities of the recombinant strain harboring fused enzyme CBM68-BLA702 showed a relative highest value of 1248 U/mL (Figure 1b). When the same enzyme BLA702 was guided by various signal peptides, such as its original signal peptide (SP), the endogenous signal peptides of *B. subtilis* LipA and YncM showed extracellular amylase activities of less than 250 U/mL (Figure 1b). In addition, the codon-optimized gene guided using the signal peptide LipA also showed a relatively low extracellular amylase activity (Figure 1b). Thus, we concluded that the N-terminal CBM68 domain played a vital role in enhancing the secretory expression of BLA702 in *B. subtilis*. Moreover, through SDS-PAGE determination of the extracellular proteins of the recombinant strains, all strains were shown to express the target enzymes at 55 kDa, using the original protein produced by the wild-type strain *B. licheniformis* 702 as a control (Figure 1c). However, the recombinant strain with the fused enzyme CBM68-BLA702 showed two target recombinant enzymes in the extracellular proteins after 48 h fermentation: one was at 68 kDa, which may have been the whole fused enzyme CBM68-BLA702, and the other was BLA702, which was at 55 kDa (Figure 1c). This indicated that the CBM68 domain may separate itself from CBM68-BLA702 in the extracellular environment of *B. subtilis*. To confirm this, we monitored the growth, extracellular amylase activity, and extracellular proteins produced during a 84 h fermentation of *B. subtilis* SCK6 (pMAC68-BLA702). The results showed that extracellular CBM68-BLA702 and BLA702 were detected after 14 h of fermentation (Figures S2 and S3b). The expression of the two target proteins increased gradually between 14 and 48 h of fermentation (Figure S2). However, after 48 h of fermentation, the expression of CBM68-BLA702 was decreasing gradually, while the expression of BLA702 continued to increase (Figure S2). The highest extracellular amylase activity was achieved at 80–84 h fermentation (Figure S3).

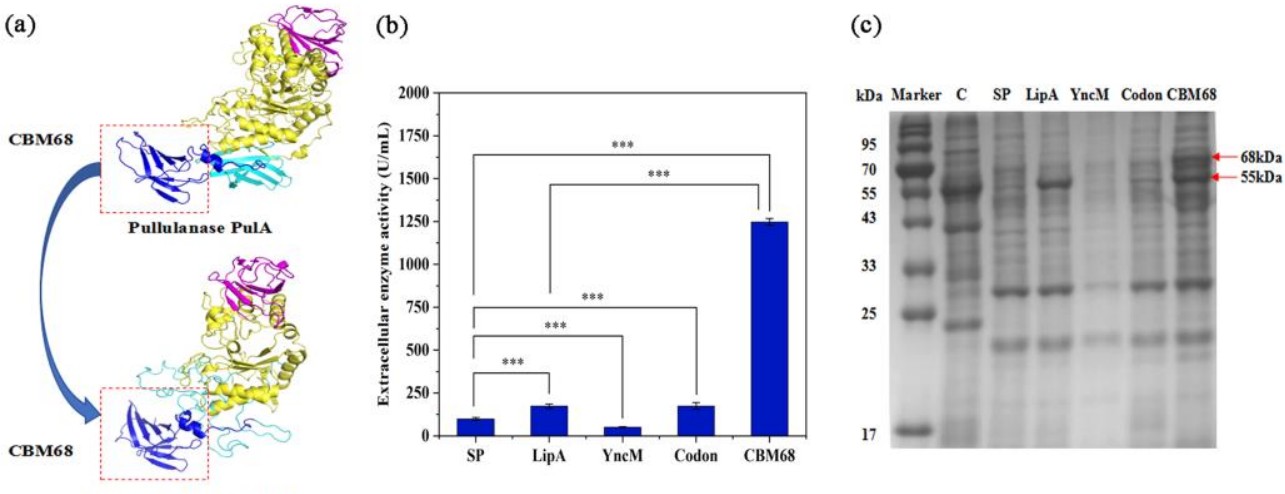

**Figure 1.** Extracellular expression of BLA702 guided by various N-terminal signal peptides in *B. subtilis*. (**a**) Comparison of the three-dimensional structure between pullulanase (PulA) and the novel fused enzyme CBM68-BLA702; (**b**) Extracellular amylase activities of the recombinant strains guided by different signal peptides; All values are expressed as means $\pm$ SDs ($n$ = 3); *** means $p < 0.001$; (**c**) SDS-PAGE of the extracellular proteins produced by the recombinant strains guided by different signal peptides.

### 3.2. Macromolecular Protein Transporters Effecting the Secretory Expression of BLA702 in *B. subtilis* SCK6

In the recombinant strain *B. subtilis* SCK6 (pMAC68-BLA702), 51 macromolecular protein transporters were selected using software analysis (Table S2). Each of the transporters was knocked out from the genome of *B. subtilis* SCK6 (pMAC68-BLA702), to determine the effects on the secretory expression of BLA702. As seen in the results in Figure 2a, four transporters (0127, 0059, 4127, and 3624) enhanced the extracellular amylase activity by more than 1.8 times when knocked out from the genome of *B. subtilis* SCK6 (pMAC68-BLA702). Among these, knocking out *ytxE* (0127) enhanced the extracellular amylase activity the most (2.5 times). Moreover, five transporters (0046, 0572, 2143, 1951, and 1521) obviously decreased the extracellular amylase activity through silent expression (Figure 2a). Through overexpressing each of the five transporters in *B. subtilis* SCK6 (pMAC68-BLA702), 0572 and 2143 enhanced the extracellular amylase activity by 1.62 and 1.34 times, respectively. Meanwhile, the other three transporters did not enhance the expression of BLA702 (Figure 2b). Target protein expression showed a positive correlation with the extracellular amylase activity (Figure 2b,c). Thus, the expressions of *comEA* (0572) and *ytxE* (0127) were positively and negatively correlated, respectively, with the extracellular secretory expression of BLA702 in *B. subtilis* SCK6 (pMAC68-BLA702).

### 3.3. Genome Integration Expression of BLA702 in *B. subtilis* SCK6

In order to construct a production strain ofBLA702 with more stable expression, we chose to integrate the *BLA702* gene into the *amyE* and *nprB* sites of the genome of *B. subtilis* SCK6. As shown in Figure 3a, different integration positions resulted in different extracellular amylase activities. Moreover, having two copies in the two positions resulted a relatively lower extracellular amylase activity than a single copy in one position (Figure 3a). Through detecting the protein expressions in both extracellular and intracellular inclusion bodies, we found that the double-copy recombinant strain formed many inclusion bodies after 48 h of fermentation (Figure 3b). Thus, a molecular chaperone protein PrsA was overexpressed in the double-copy recombinant strain, to facilitate the soluble expression of BLA702. The results showed that the inclusion bodies decreased remarkably by overexpressing PrsA,

and the extracellular enzyme activity increased to 1230.5 U/mL after 48 h of fermentation (Figure 3).

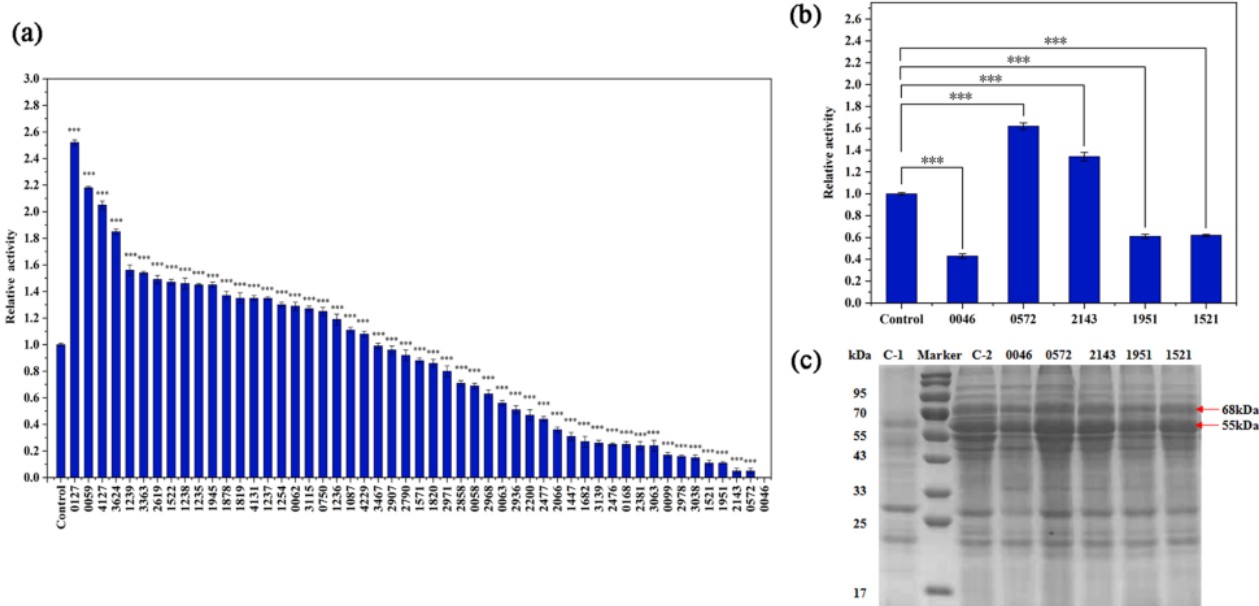

**Figure 2.** Effects of different macromolecular protein transporters on the extracellular amylase activity. (**a**) Effects of knocking out each of the 51 macromolecular protein transporters; all values are expressed as means ± SDs ($n = 3$); "***" means $p < 0.001$; (**b**) Effects of overexpression of the key transporters; all values are expressed as means ± SDs ($n = 3$); *** means $p < 0.001$; (**c**) SDS-PAGE of the extracellular proteins of the strains overexpressing the key transporters. C-1 means the negative control strain *B. subtilis* SCK6 (pMAC68); C-2 means the positive control strain *B. subtilis* SCK6 (pMAC68-BLA702).

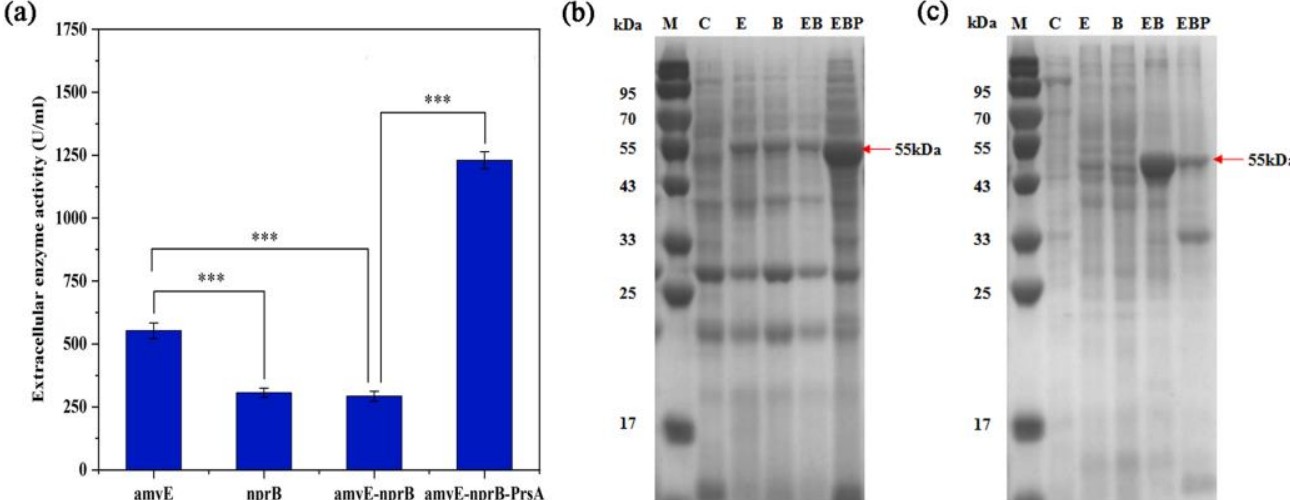

**Figure 3.** Expression of BLA702 by different genome integration strains. (**a**) Extracellular amylase activities of different genome integration strains; all values are expressed as means ± SDs ($n = 3$); *** means $p < 0.001$; (**b**) SDS-PAGE of the extracellular proteins of different genome integration strains; (**c**) SDS-PAGE of the inclusion bodies of different genome integration strains. M signifies the molecular weight standards of proteins; C signifies the control strain *B. subtilis* SCK6; E signifies the genome integration strain *B. subtilis* SCK6 (AmyE::BLA702); B signifies the genome integration strain *B. subtilis* SCK6 (NprB::BLA702); EB signifies the genome integration strain *B. subtilis* SCK6 (AmyE::BLA702-NprB::BLA702); EBP signifies the genome integration strain *B. subtilis* SCK6 (AmyE::BLA702-NprB::BLA702-PrsA).

　　　On the above basis, multiple strategies for increasing the secretory expression of BLA702 in *B. subtilis* SCK6 were employed in the construction of genome integration strains. The inactivation of *ytxE* could slightly increase the extracellular amylase activity and reduced inclusion body formation by a certain amount (Figure 4a,c). On this basis, adding the CBM68 domain in the N-terminal remarkably enhanced the extracellular activity and protein expression (Figure 4a,b). However, quite a few inclusion bodies also formed in the cells (Figure 4c). To further decrease the inclusion body formation, *comEA* (0572) and PrsA were overexpressed. Both of them have the function of reducing inclusion body formation and enhancing the extracellular soluble expression of BLA702 (Figure 4). By comparison, the overexpression of PrsA had more obvious effects on enhancing the extracellular soluble expression of BLA702. The final recombinant strain *B. subtilis* 0127 (AmyE::BLA702-NprB::CBM68-BLA702-PrsA) produced the highest extracellular amylase activity at 2244.3 U/mL after 48 h of fermentation (Figure 4a).

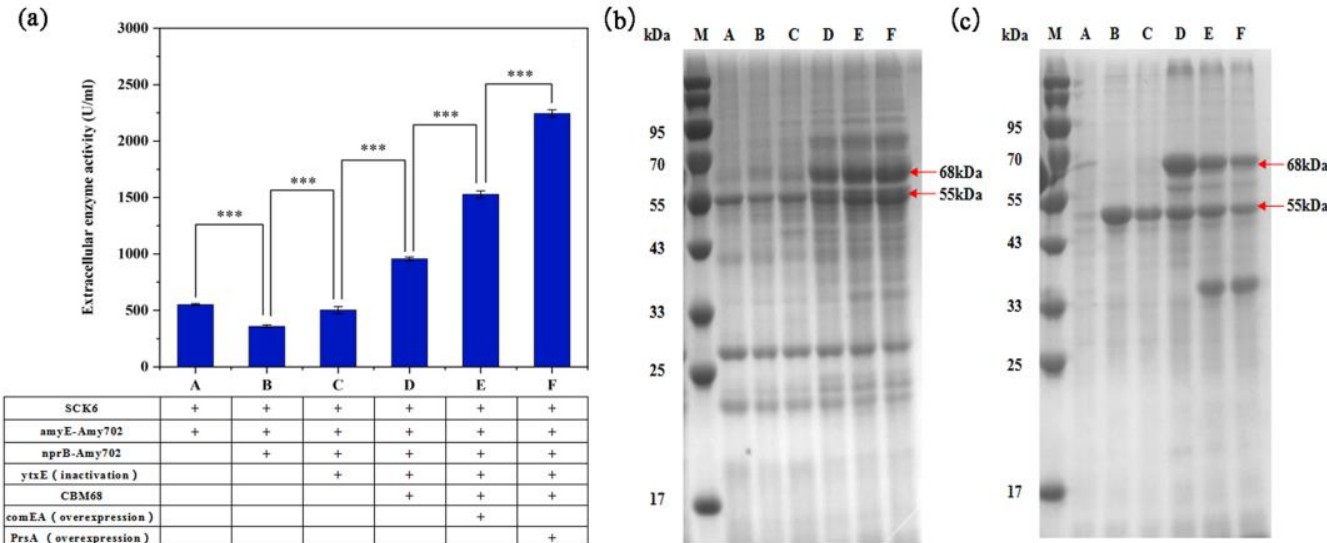

**Figure 4.** Different strategies for increasing the secretory expression of BLA702 in *B. subtilis* SCK6. (**a**) Extracellular amylase activity of different genome integration strains; A–F were the genome integration strains constructed using step-by-step increase strategies to improve the protein expression or secretion from the original single site (amyE) integration strain (A); SCK6 combined with amyE-BLA702 signifies integrating the BLA702 gene into the amyE site of the genome of *B. subtilis* SCK6; nprB-BLA702 involved adding one copy of BLA702 gene at the nprB site of the genome at the basis; *ytxE* (inactivation) involved silencing the *ytxE* gene of *B. subtilis* SCK6 to form *B. subtilis* 0127; CBM68 involved adding the CBM68 domain on the N-terminal of BLA702, which was integrated into the nprB site; *comEA* (overexpression) involved integrating the *comEA* gene into the *aprE* gene site of the genome of *B. subtilis* 0127 on the above basis; similarly, PrsA (overexpression) involved integrating the PrsA encoding gene into the *aprE* gene site of the genome of *B. subtilis* 0127 on the above basis; all values are expressed as means $\pm$ SDs ($n = 3$); *** means $p < 0.001$; (**b**) SDS-PAGE of the extracellular proteins of different genome integration strains; (**c**) SDS-PAGE of the inclusion bodies of different genome integration strains.

### 3.4. BLA702 Production and Strain Stability of the Engineered Strains

　　　To further compare the enzyme production capacity of the genome integration strain *B. subtilis* 0127 (AmyE::BLA702-NprB::CBM68-BLA702-PrsA) and the plasmid expression strain *B. subtilis* 0127 (pMAC68-BLA702), the two strains were fermented in a 500 mL shake flask for 168 h. As shown in Figure 5, *B. subtilis* 0127 (AmyE::BLA702-NprB::CBM68-BLA702-PrsA) produced the highest extracellular amylase activity of 3745.7 U/mL at 126 h fermentation, while *B. subtilis* 0127 (pMAC68-BLA702) produced the highest extracellular amylase activity of 3825.4 U/mL at 102 h fermentation (Figure 5). This indicated

that *B. subtilis* 0127 (AmyE::BLA702-NprB::CBM68-BLA702-PrsA) had the same enzyme production capacity as *B. subtilis* 0127 (pMAC68-BLA702) but needed a slightly longer fermentation time.

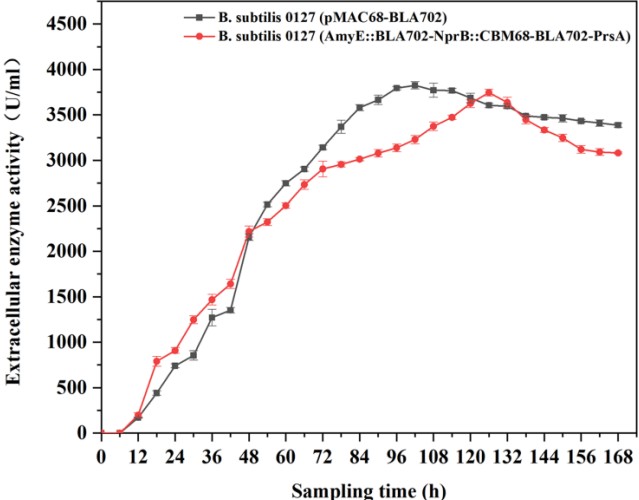

**Figure 5.** Profile of the extracellular amylase activity of *B. subtilis* 0127 (pMAC68-BLA702) and *B. subtilis* 0127 (AmyE::BLA702-NprB::CBM68-BLA702-PrsA). All values are expressed as means $\pm$ SDs (*n* = 3).

After 10 generations of successive inoculation, without adding antibiotics, the single colonies of *B. subtilis* 0127 (pMAC68-BLA702) showed a remarkable reduction (5.5–21.6%) in the capacity of their enzyme production (Figure 6a), while the single colonies of *B. subtilis* 0127 (AmyE::BLA702-NprB::CBM68-BLA702-PrsA) showed no difference in their capacity for enzyme production with the original strain (Figure 6b). This indicated that the genome integration strain *B. subtilis* 0127 (AmyE::BLA702-NprB::CBM68-BLA702-PrsA) is relatively more suitable for industrial production of BLA702.

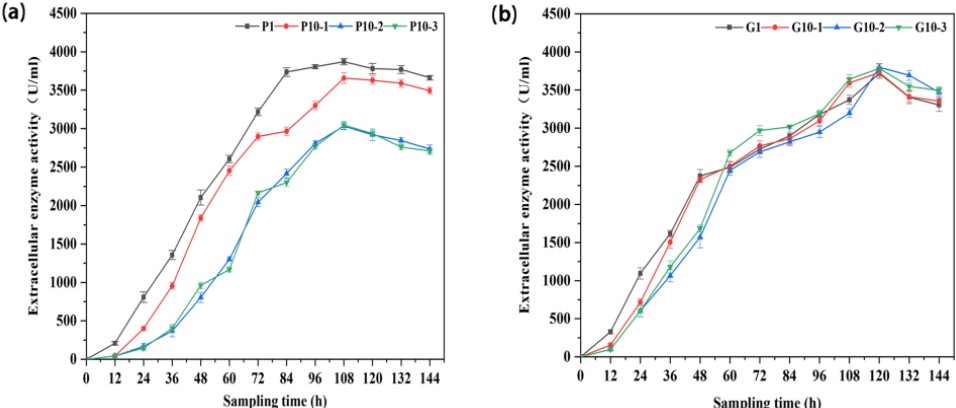

**Figure 6.** Stability of passage of *B. subtilis* 0127 (pMAC68-BLA702) and *B. subtilis* 0127 (AmyE::BLA702-NprB::CBM68-BLA702-PrsA). (**a**) Profile of extracellular amylase activity of *B. subtilis* 0127 (pMAC68-BLA702) after 10 generations of successive inoculation without adding antibiotics. P1 signifies the original strain, P10-1∼P10-3 signify the single colonies of *B. subtilis* 0127 (pMAC68-BLA702) after 10 generations of successive inoculation without adding antibiotics; (**b**) profile of the extracellular amylase activity of *B. subtilis* 0127 (AmyE::BLA702-NprB::CBM68-BLA702-PrsA) after 10 generations of successive inoculation without adding antibiotics. G1 signifies the original strain, G10-1∼G10-3 signify the single colonies of *B. subtilis* 0127 (AmyE::BLA702-NprB::CBM68-BLA702-PrsA) after 10 generations of successive inoculation without adding antibiotics. All values are expressed as means $\pm$ SDs (*n* = 3).

## 4. Discussion

In this study, we proved that the N-terminal CBM68 domain could enhance the secretory expression of BLA702 in *B. subtilis* through a non-classical secretion pathway. In a signal peptide prediction analysis on the N-terminal of CBM68, no typical signal peptide sequence was found [9]. Thus, we chose to use the whole domain of CBM68 in the secretory expression of BLA702. A length of the loop was reserved for the connection with the N-terminal of BLA702, avoiding effects on the activity of the fused enzyme (Figure 1a). However, the results showed that a part of the fused enzymes (CBM68-BLA702) was truncated to BLA702 in the broth, based on detecting the molecular weight of the target proteins (Figure 1c). Moreover, with the extension of the fermentation time, more and more CBM68-BLA702 was truncated to BLA702, and after 84 h of fermentation, there was no CBM68-BLA702 left in the broth (Figures S2 and S4). According to the increase in the curve of the extracellular amylase activity during fermentation, the truncated BLA702 was the main source of enzyme activity (Figures S2 and S3). However, the specific break location of the fusion enzyme remains to be determined. or now, according to the results of this study, CBM68 at the N-terminal of BLA702 was shown to remarkably enhance the secretory expression of BLA702 in *B. subtilis*. Compared with the original signal peptide (SP) and the high secretion signal peptide LipA, the extracellular activity of BLA702 guided by CBM68 was up to 1248 U/mL, which was 12.6 and 7.2 times higher than those guided by the two signal peptides, respectively (Figure 1b).

As CBM68 led the proteins being secreted to the extracellular through a non-classical secretion pathway, there is insufficient information on the transporters related to the novel secretory pathway at present. Thus, we constructed a single gene knockout strain library containing 51 genes encoding the macromolecular transporters, to detect the effect of each transporter on the secretory expression of CBM68-BLA702. The results showed that most of the transporters affected the secretory expression of CBM68-BLA702 to varying degrees (Figure 2a). This indicated that CBM68 was transported to the extracellular through a complex transporter system, and many macromolecular transporters are required to participate in its transport. Among these transporters, we identified two key transporters, 0572 and 2143, whose knocking out almost stopped the secretory expression of CBM68-BLA702 and whose overexpression remarkably enhanced the secretory expression of CBM68-BLA702 (Figure 2). The gene *comEA* (0572) encodes a membrane-bound high-affinity DNA-binding receptor. The gene *spoIIQ* (2143) encodes a stage II sporulation protein. Both of these transporters may be key factors in the transport of CBM68-BLA702. Moreover, several transporters obviously obstructed the secretory expression of CBM68-BLA702, such as 0127, 0059, 4127, and 3624 (Figure 2a). The gene *ytxE* (0127) encodes an ABC transporter ATP-binding protein. The gene *ytrF* (0059) encodes an ABC transporter permease. The gene *trpP* (4127) encodes a tryptophan transporter. The gene *gabP* (3624) encodes a GABA permease. This indicated that the ABC transporters and some specific amino acid transporters may compete with the protein transport channels of CBM68-BLA702.

The single gene silencing strain *B. subtilis* 0127 (pMAC68-BLA702) showed a 2.5 times higher secretory expression of BL702 than *B. subtilis* SCK6 (pMAC68-BLA702) (Figure 2a). It could produce more than 2741.9 U/mL extracellular amylase after 48 h fermentation. However, the recombinant strain relies on the addition of antibiotics to stably maintain a high number of plasmids copies, which is not suitable for industrial production. Therefore, we attempted to construct genome integration strains with a high extracellular expression level in this study. The results showed that a single copy genome integration strain could only produce a maximum 552.6 U/mL of extracellular amylase (Figure 3a). To further increase the expression level of the target protein, a double-copy genome integration strain *B. subtilis* SCK6 (AmyE::BLA702-NprB::BLA702) was constructed. However, the extracellular amylase activity of *B. subtilis* SCK6 (AmyE::BLA702-NprB::BLA702) was lower than that of the single-copy genome integration strains, due to the formation of inclusion bodies (Figures 3 and 4). The single gene silencing strain *B. subtilis* 0127 (AmyE::BLA702-NprB::BLA702) had a 40.1% higher extracellular activity than *B. subtilis* SCK6 (AmyE::BLA702-NprB::BLA702)

but was still lower than 500 U/mL (Figure 4a). When the CBM68 domain was added to one copy of BLA702, to construct the recombinant strain *B. subtilis* 0127 (AmyE::BLA702-NprB::CBM68-BLA702), the extracellular amylase activity was up to 955.8 U/mL, which was 2.7 times higher than that of *B. subtilis* SCK6 (AmyE::BLA702-NprB::BLA702). However, although CBM68 could enhance the extracellular expression of BLA702, there was still a certain amount of inclusion bodies in the cells (Figure 4c). Thus, the CBM68 domain was more suitable for adding one copy of BLA702 at a relatively lower expression level position at NprB (Figure 3a). To further decrease the formation of inclusion bodies, the two strains *B. subtilis* 0127 (AmyE::BLA702-NprB::CBM68-BLA702-*oxaAB*) and *B. subtilis* 0127 (AmyE::BLA702-NprB::CBM68-BLA702-PrsA) were constructed, with a 4.3 times and 6.2 times enhancement in their extracellular amylase activity, respectively (Figure 4a). *B. subtilis* 0127 (AmyE::BLA702-NprB::CBM68-BLA702-PrsA) showed the lowest inclusion bodies formation (Figure 4c). Compared with *B. subtilis* 0127 (pMAC68-BLA702), the genome integration strain *B. subtilis* 0127 (AmyE::BLA702-NprB::CBM68-BLA702-PrsA) showed a similar extracellular expression level (3745.7 U/mL) after 126 h fermentation and a better stability of passage (Figures 5 and 6). Therefore, the newly constructed strain *B. subtilis* 0127 (AmyE::BLA702-NprB::CBM68BLA702-PrsA) has potential as an industrial production strain for BLA702. Moreover, the strategies used in this work, especially the usage of CBM68 and PrsA, could provide a reference for the extracellular soluble expression of other foreign proteins in *Bacillus subtilis*.

**Supplementary Materials:** The following supporting information can be downloaded at https://www.mdpi.com/article/10.3390/fermentation9020139/s1, Table S1: Primers used in this study; Table S2: Macromolecular protein transporters in this study; Table S3: The N20 sequences of the macromolecular protein transporters; Figure S1: Schematic diagrams of plasmid construction; Figure S2: SDS-PAGE of the extracellular proteins of *B. subtilis* SCK6 (pMAC68-BLA702); Figure S3: Growth profile (a) and extracellular amylase activity profile (b) of *B. subtilis* SCK6 (pMAC68-BLA702); Figure S4: SDS-PAGE of the extracellular proteins of *B. subtilis* 0127 (pMAC68-BLA702) and *B. subtilis* 0127 (AmyE::BLA702-NprB::CBM68-BLA702-PrsA).

**Author Contributions:** Conceptualization, H.Z. and H.S.; methodology, X.F. and Y.Y.; software, H.Z.; validation, X.F., Y.Y. and J.X.; formal analysis, X.Z.; investigation, X.Z. and J.X.; resources, H.Z.; data curation, Y.Y.; writing—original draft preparation, H.Z.; writing—review and editing, H.Z. and Y.L.; visualization, Y.Y.; supervision, W.B. and H.S.; project administration, H.Z.; funding acquisition, H.Z. and H.S. All authors have read and agreed to the published version of the manuscript.

**Funding:** This research was funded by the State Key Research and Development Program of China, grant number 2021YFC2100405, 2021YFC2100201, and 2021YFC2100403 and Tianjin Synthetic Biotechnology Innovation Capacity Improvement Project, grant number TSBICIP-PTJJ-007-13 and TSBICIP-KJGG-009-0202.

**Data Availability Statement:** The data presented in this study are available in the text of the article and the Supplementary Materials, and further inquiries can be directed to the corresponding author.

**Acknowledgments:** We thank Meng Wang from Tianjin Institute of Industrial Biotechnology, Chinese Academy of Sciences for constructing the single gene knockout strain library.

**Conflicts of Interest:** The authors declare no conflict of interest. The funders had no role in the design of the study; in the collection, analyses, or interpretation of data; in the writing of the manuscript; or in the decision to publish the results.

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
