# Peer review of "Overexpression of a Thermostable α-Amylase through Genome Integration in Bacillus subtilis"

_fermentation, doi:10.3390/fermentation9020139_

Round 1
Reviewer 1 Report
In the present study Yang et al overexpressed thermostable α-amylase from Bacillus licheniformis 702 in Bacillus 21 subtilis using a genome integration approach and this enzyme was guided to extracellular environment by number of signal peptide. This were further improved by expressed it with the best transporters and chaperone protein. The study is interesting however the paper is not well written
My concerns,
The paper lacks rigorous statistical analysis, though authors claimed in the methods that they performed statistical analysis these analysis are nowhere shown in the result section.
I recommend an extensive English editing by a Native English to the present manuscript before any acceptance can be made,
please move some of supplementary materials e.g, primers, oligo and guide RNA used for knockout into the methodology of the manuscript
Please label the protein markers with their molecular size in every gel reported in the manuscript
Please move the figures label e.g., a, b, c, etc above the figures not down.
In figure 2b, whats the unit of extracellular enzyme activity?
Author Response
Response to the reviewer’s comments:
Reviewer: 1
In the present study Yang et al overexpressed thermostable α-amylase from Bacillus licheniformis 702 in Bacillus 21 subtilis using a genome integration approach and this enzyme was guided to extracellular environment by number of signal peptide. This were further improved by expressed it with the best transporters and chaperone protein. The study is interesting however the paper is not well written
My concerns,
The paper lacks rigorous statistical analysis, though authors claimed in the methods that they performed statistical analysis these analysis are nowhere shown in the result section.
Response: Thanks for the reviewer’s kind reminder. The statistical analyzed results on the significance of different data have been added in every figure in the revised manuscript. The correspondent clarifications have also been added in the figure legends.
I recommend an extensive English editing by a Native English to the present manuscript before any acceptance can be made,
Response: A carefully English editing of the whole manuscript has been made by a native English-speaking colleague.
please move some of supplementary materials e.g, primers, oligo and guide RNA used for knockout into the methodology of the manuscript
Response: We have moved the table S1 of supplementary materials which listed plasmids used in the study to the text as table 1 of the revised manuscript according to the reviewer’s kind suggestion. Other tables in supplementary materials were too big to list in the text of the paper.
Please label the protein markers with their molecular size in every gel reported in the manuscript
Response: The protein markers have been labeled with their molecular sizes in every SDS-PAGE picture in the revised manuscript.
Please move the figures label e.g., a, b, c, etc above the figures not down.
Response: All of the figures have been modified according to the reviewer’s requirement.
In figure 2b, whats the unit of extracellular enzyme activity?
Response: Thanks for the reviewer’s kind reminder. It has been corrected.
Reviewer 2 Report
Overly, it seems that this study is new and novel. This paper was well-designed and can be accepted after some modifications and revisions. Anyway, the author can find some comments in the following.
The interval used for the measurement of extracellular amylase activity and protein expression should be mentioned in the M&M.
It would be better to illustrate the constructed plasmid in a figure.
The kDa of the protein ladder should be written in all the figures.
In Fig 2a, the macromolecular protein transporters did not clear.
In Fig 3 b, c, the molecular weight of M and the band of interest should be illustrated.
In Fig 4a, the Extracellular amylase activity of different genome integration strains, A-F, and the table should be clarified in the caption. Also, the table was not clear; please increase the font size.
Author Response
Response to the reviewer’s comments:
Reviewer: 2
Overly, it seems that this study is new and novel. This paper was well-designed and can be accepted after some modifications and revisions. Anyway, the author can find some comments in the following.
The interval used for the measurement of extracellular amylase activity and protein expression should be mentioned in the M&M.
Response: Thanks for the reviewer’s kind suggestion. The relevant exposition has been added in the section “2.6 Determination of the extracellular amylase activity and protein expression” in the revised manuscript.
It would be better to illustrate the constructed plasmid in a figure.
Response: Three figures illustrating the plasmids construction have been provided in the supplementary materials as Figure S1 according to the reviewer’s kind suggestion. It has also been cited in the text of the revised manuscript.
The kDa of the protein ladder should be written in all the figures.
Response: The protein markers have been labeled with their molecular sizes in every SDS-PAGE picture in the revised manuscript.
In Fig 2a, the macromolecular protein transporters did not clear.
Response: We have improved the resolution of every figure in the revised manuscript. The high-resolution images have been also uploaded separately with the revised manuscript.
In Fig 3 b, c, the molecular weight of M and the band of interest should be illustrated.
Response: The pictures have been modified according to the reviewer’s requirement.
In Fig 4a, the Extracellular amylase activity of different genome integration strains, A-F, and the table should be clarified in the caption. Also, the table was not clear; please increase the font size.
Response: Supplemented clarifications have been added in the figure caption according to the reviewer’s suggestion. The resolution of the picture has also been improved and also been uploaded separately with the revised manuscript.
Round 2
Reviewer 1 Report
The authors have addressed several issues however, some figures are distorted and others figure ie 5 and 6 lack statistical analysis.
Author Response
The authors have addressed several issues however, some figures are distorted and others figure ie 5 and 6 lack statistical analysis.
Response: Thanks for the reviewer’s comments. All of the figures have been improved according to the“instruction for authors” which with the resolutions higher than 600 dpi and with >1000 pixels at both width and height. The improved figures have been submitted in a single zip archive during submission of the revised manuscript. As to Figure 5 and Figure 6, each data in the curve is the mean value of three repilcations with the error bar. The difference in data on every point of the curve is also statistically significant. The relative clarifications have been supplemented in the figure captions of Figure 5 and Figure 6.